# Data Augmentation with Diffusion for Open-Set Semi-Supervised Learning

**Seonghyun Ban**[1*],   **Heesan Kong**[1*],   **Kee-Eung Kim**[1, 2†]

[1]Kim Jaechul Graduate School of AI, KAIST
[2]School of Computing, KAIST
shban@ai.kaist.ac.kr, hskong@ai.kaist.ac.kr, kekim@kaist.ac.kr

## Abstract

Semi-supervised learning (SSL) seeks to utilize unlabeled data to overcome the limited amount of labeled data and improve model performance. However, many SSL methods typically struggle in real-world scenarios, particularly when there is a large number of irrelevant instances in the unlabeled data that do not belong to any class in the labeled data. Previous approaches often downweight instances from irrelevant classes to mitigate the negative impact of class distribution mismatch on model training. However, by discarding irrelevant instances, they may result in the loss of valuable information such as invariance, regularity, and diversity within the data. In this paper, we propose a data-centric generative augmentation approach that leverages a diffusion model to enrich labeled data using both labeled and unlabeled samples. A key challenge is extracting the diversity inherent in the unlabeled data while mitigating the generation of samples irrelevant to the labeled data. To tackle this issue, we combine diffusion model training with a discriminator that identifies and reduces the impact of irrelevant instances. We also demonstrate that such a trained diffusion model can even convert an irrelevant instance into a relevant one, yielding highly effective synthetic data for training. Through a comprehensive suite of experiments, we show that our data augmentation approach significantly enhances the performance of SSL methods, especially in the presence of class distribution mismatch.

## 1   Introduction

Deep neural networks (DNNs), trained using a large amount of labeled datasets, have shown to achieve remarkable performance in a variety of supervised learning tasks, such as image classification (LeCun et al., 2015; Krizhevsky et al., 2017) and object detection (Everingham et al., 2010; Lin et al., 2014). Nonetheless, the intensive labor of annotating vast datasets often renders the construction of sufficiently large labeled datasets prohibitively expensive for numerous applications (Oliver et al., 2018). To address this, semi-supervised learning (SSL) (Chapelle et al., 2009) has emerged as a viable approach, which aims to leverage abundant unlabeled data to overcome the limited availability of labeled data.

Recent progress in SSL has made many noteworthy advancements, including techniques such as pseudo-labeling (Lee, 2013; Pham et al., 2021), consistency regularization (Sajjadi et al., 2016; Tarvainen & Valpola, 2017; Sohn et al., 2020), and entropy minimization (Grandvalet & Bengio, 2004; Miyato et al., 2018; Berthelot et al., 2019). However, a common limitation of these methods is their reliance on the critical assumption that both unlabeled and labeled data instances are drawn

---

[*]Equal Contribution.
[†]Corresponding Author.

38th Conference on Neural Information Processing Systems (NeurIPS 2024).

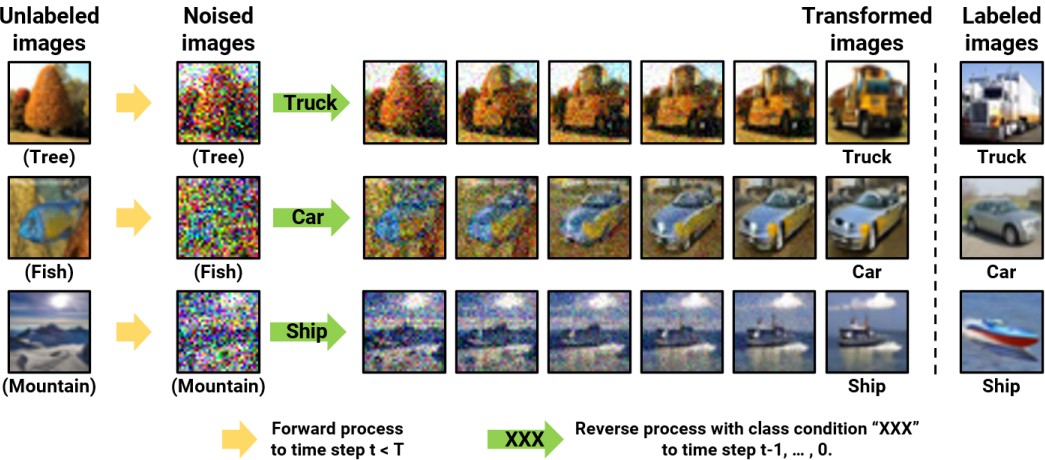

Figure 1: Transforming unlabeled data using a diffusion model. Initially, the unlabeled data includes classes like trees, fish, and mountains, which are irrelevant to the labeled data's classes such as trucks, cars, and ships. The reverse process with class conditioning resolves this mismatch while preserving the diversity of the original unlabeled samples. More examples can be found in Appendix H.

from an identical distribution. This often leads to significant performance degradation when there is a class distribution mismatch, as noted in Oliver et al. (2018). Given that real-world settings often deviate from this assumption (Guo et al., 2020), it becomes crucial to address the class distribution mismatch for successfully applying SSL in realistic scenarios.

A prevalent approach to mitigate the class distribution mismatch in SSL involves selectively utilizing unlabeled data through a weighting function which serves to reduce the influence of irrelevant unlabeled samples (Chen et al., 2020; Guo et al., 2020). While these filtering methods are intuitively appealing and have shown their effectiveness in mitigating negative effect of the class distribution mismatch, they may result in the loss of valuable information such as invariance, regularity, and diversity within the data: even though the labeled data contains only white trucks, could we generate images of red trucks using the red bike images in the unlabeled dataset?

In this paper, we propose a generative data augmentation approach that leverages a diffusion model to enrich labeled data using both labeled and unlabeled samples. A key challenge lies in utilizing the diversity of the unlabeled data to compensate the limited amount of labeled data, while minimizing the generation of samples that are irrelevant to the classes in the labeled dataset. To address this, we integrate diffusion model training with a discriminator that evaluates the relevance of each unlabeled instance. The discriminator's resulting score is used to assign weights to unlabeled instances, allowing those with higher relevance to contribute more significantly to the training of the diffusion model.

In addition, drawing inspiration from the approach in Meng et al. (2022), we add noise to each unlabeled sample and utilize them as guide images during the data generation process. As depicted in Figure 1, we found that incorporating class conditions into this generation process can transform possibly irrelevant unlabeled samples into labeled samples while preserving key characteristics of the original unlabeled samples (e.g., outline, color arrangement, shape, etc.).

Our extensive experimental results, utilizing CIFAR-10, CIFAR-100 (Krizhevsky & Hinton, 2009), ImageNet-30 (Deng et al., 2009), and ImageNet-100 (Cao et al., 2022) datasets with six baseline methods, demonstrate that our approach further improves the performance of recent SSL methods, especially under the class distribution mismatch.

## 2 Related Works

### 2.1 Standard Semi-Supervised Learning

Semi-Supervised Learning (SSL) aims to leverage both labeled and unlabeled data to mitigate issues related to the scarcity and high annotation cost of labeled data. Since many SSL methods operate under the assumption that labeled and unlabeled data are sampled from an identical distribution,

we refer to this as the standard SSL setting. Among the various approaches to the standard SSL setting, we briefly review two of the most representative methods: pseudo-labeling and consistency regularization.

Pseudo-labeling (Scudder, 1965; McLachlan, 1975; Lee, 2013) is a technique where the model-predicted labels of the unlabeled data are treated as if they were true labels. Essentially, this method simply converts unlabeled data into labeled data. On the other hand, consistency regularization (Bachman et al., 2014) has become a crucial component in recent SSL regimes. This approach applies data augmentation on the unlabeled data to regularize the model to yield similar outputs for augmented views of the same instance (Sajjadi et al., 2016; Laine & Aila, 2017; Sohn et al., 2020).

However, these methods often fail when there is a mismatch between the distributions of labeled and unlabeled data. In some cases, their performance may be even worse than simply discarding all the unlabeled data (Oliver et al., 2018). The primary source of the problem is the presence of unlabeled instances that do not belong to any of the classes present in the labeled data, which we refer to as out-of-distribution (OOD) instances. They may exacerbate confirmation bias (Arazo et al., 2020) in pseudo-labeling approaches or intensify the overconfidence problem in consistency regularization approaches (Chen et al., 2020).

## 2.2 Open-set Semi-Supervised Learning

The open-set SSL setting refers to the practical yet challenging scenario where the class distributions of labeled and unlabeled dataset differ significantly. A prevalent solution to the class distribution mismatch is to filter out OOD instances from the unlabeled dataset. To achieve this, it is necessary to accurately identify them, despite the absence of label information. Uncertainty-Aware Self-Distillation (UASD), proposed by Chen et al. (2020), formulates temporally ensembled networks and utilizes ensemble prediction to quantify predictive uncertainty of labels to identify OOD instances. DS3L (Guo et al., 2020) adopts a meta-learning approach to selectively use unlabeled data that enhances generalization performance. OpenMatch (Saito et al., 2021) leverages one-vs-all classifiers as the OOD detector to filter out OOD instances. Safe-Student (He et al., 2022) employs teacher-student mechanism and introduces energy-discrepancy, a new scoring function for detecting OOD instances. The above methods follow the detect-and-filter paradigm, which is the dominant approach of open-set SSL, assuming that OOD instances are fundamentally harmful.

In contrast to these detect-and-filter approaches, several studies share similar goals to ours, aiming to harness the potential of OOD unlabeled data rather than simply discarding them. T2T (Huang et al., 2021) incorporates a warm-up training step using OOD instances to perform a self-supervised pretext task for learning effective discriminative features. TOOR (Huang et al., 2022) introduces a weighting mechanism to evaluate the transferability of each OOD instance based on domain similarity and class tendency, and uses adversarial domain adaptation to align the feature distributions of transferable OOD instances and in-distribution (ID) instances. Fix-A-Step (Huang et al., 2023) leverages OOD instances to obtain useful data augmentation to promote diversity of training data, and integrate this idea into MixMatch (Berthelot et al., 2019). IOMatch (Li et al., 2023) takes into consideration that the OOD detector may be unreliable, particularly when labeled data are scarce. It instead employs a multi-binary classifier to produce unified open-set pseudo-labels for labeled and unlabeled data, including OOD instances.

Our approach significantly deviates from these open-set SSL methods. It is primarily centered on developing an effective data augmentation strategy from both labeled and unlabeled data, which can be used to transform an unlabeled OOD instance into a labeled ID instance.

## 2.3 Recent Diffusion-based Augmentation Approaches

In this section, we provide a detailed comparison of DWD with diffusion-based methods, namely DPT (You et al., 2023) and DA-Fusion[1] (Trabucco et al., 2024).

DPT is a simple yet effective method that integrates diffusion models into SSL. To generate semantically accurate images, they trained a semi-supervised classifier on partially labeled real images and used it to assign pseudo-labels for all the data. Subsequently, they trained a conditional diffusion

---

[1]Although DA-Fusion is not a SSL algorithm, we include it as a supervised learning baseline due to its methodological similarity in utilizing image-to-image generation process.

model using the pseudo-labels to synthesize images and then re-trained the semi-supervised classifier using these generated images. However, they still rely on the fundamental assumption behind the standard SSL setting. When faced with mismatch in class distributions, they suffer from the confirmation bias in pseudo labels of OOD unlabeled images and become ineffective. In contrast, our approach addresses the class distribution mismatch by integrating the discriminator into the training of the diffusion model.

DA-Fusion is a data augmentation method that utilizes a large pretrained text-to-image diffusion model (i.e., Stable Diffusion, Rombach et al., 2022). Similar to ours, it also initiates the reverse process with partially noised real images rather than generating images from scratch. However, the purpose of the generation process is distinctly different from that of our approach. While DA-Fusion aims to augment given labeled samples with subtle visual details already contained in the pretrained diffusion model, our method properly trains a diffusion model to capture both the labeled data distribution and the diversity of unlabeled samples from the given datasets, and transforms irrelevant unlabeled samples into labeled ones.

## 3   Preliminaries

**Diffusion models (Sohl-Dickstein et al., 2015)**   incrementally add Gaussian noises to the data during its forward process and progressively removes this noise during the reverse process to reconstruct the original data. Given a data point $\mathbf{x}_0$, the forward process is defined as a Markov chain that produces a sequence of noisy samples $\mathbf{x}_1, ..., \mathbf{x}_T$ according to a variance schedule $\beta_1, ..., \beta_T$:

$$q(\mathbf{x}_t|\mathbf{x}_{t-1}) := \mathcal{N}(\mathbf{x}_t; \sqrt{1-\beta_t}\mathbf{x}_{t-1}, \beta_t\mathbf{I}), \qquad q(\mathbf{x}_{1:T}|\mathbf{x}_0) := \prod_{t=1}^{T} q(\mathbf{x}_t|\mathbf{x}_{t-1}) \qquad (1)$$

The forward process exhibits a notable property in that $\mathbf{x}_t$ at any arbitrary time step $t \in \{1, ..., T\}$ can be obtained in closed form:

$$\mathbf{x}_t = \sqrt{\bar{\alpha}_t}\mathbf{x}_0 + \sqrt{(1-\bar{\alpha}_t)}\boldsymbol{\epsilon} \qquad (2)$$

where $\alpha_t = 1 - \beta_t$, $\bar{\alpha}_t = \prod_{i=1}^{t} \alpha_i$ and $\boldsymbol{\epsilon} \sim \mathcal{N}(\mathbf{0}, \mathbf{I})$. As the reverse process can be expressed using the same functional form when $\beta_t$ is sufficiently small (Feller, 1949; Sohl-Dickstein et al., 2015), the reverse process is also defined as a Markov chain with learned Gaussian transitions starting at $p(\mathbf{x}_T) = \mathcal{N}(\mathbf{x}_T, \mathbf{0}, \mathbf{I})$:

$$p_\theta(\mathbf{x}_{t-1}|\mathbf{x}_t) := \mathcal{N}(\mathbf{x}_{t-1}; \boldsymbol{\mu}_\theta(\mathbf{x}_t, t), \boldsymbol{\Sigma}_\theta(\mathbf{x}_t, t)), \qquad p_\theta(\mathbf{x}_{0:T}) := p(\mathbf{x}_T)\prod_{t=1}^{T} p_\theta(\mathbf{x}_{t-1}|\mathbf{x}_t) \quad (3)$$

**Denoising Diffusion Probabilistic Models (DDPMs) Ho et al. (2020)**   propose to use a simplified Gaussian distribution parameterization of the reverse process, which sets $\boldsymbol{\Sigma}_\theta$ to time-dependent constants and reparameterizes the mean function approximator $\boldsymbol{\mu}_\theta(\mathbf{x}_t, t)$ with noise predictor $\boldsymbol{\epsilon}_\theta(\mathbf{x}_t, t)$ which predicts $\boldsymbol{\epsilon}$ in (2). The reverse process is then learned by the following objective:

$$L_{ddpm} = \mathbb{E}_{\mathbf{x}_0, t, \boldsymbol{\epsilon}}[||\boldsymbol{\epsilon}_\theta(\mathbf{x}_t, t) - \boldsymbol{\epsilon}||_2^2] \qquad (4)$$

For the conditional DDPM, the primary modification is the incorporation of conditions $\mathbf{c}$ (such as classes or texts) as an additional input, expressed as $\boldsymbol{\epsilon}_\theta(\mathbf{x}_t, \mathbf{c}, t)$. This adaptation allows the diffusion model to take into account specific conditions or attributes during the reverse process, thereby enhancing its applicability to more targeted scenarios.

**Positive-Unlabeled (PU) Learning (Liu et al., 2002; Li & Liu, 2003; Du Plessis et al., 2015; Kiryo et al., 2017)**   is a binary classification task in a situation where negative labels are missing. It aims to train models using positive-labeled and unlabeled data to perform binary classification. The main idea involves indirectly estimating the model loss on negative samples using the class prior. Given that unlabeled data are drawn from $p^u(\mathbf{x}) = \mu\, p^+(\mathbf{x}) + (1-\mu)\, p^-(\mathbf{x})$, where $\mu = p(Y = 1)$ is the class prior and $p^+(\mathbf{x}) = p(\mathbf{x}\,|\,Y = 1)$ and $p^-(\mathbf{x}) = p(\mathbf{x}\,|\,Y = -1)$ are positive and negative class-conditional densities, the loss can be reformulated as:

$$\mathbb{E}\left[\ell(g(\mathbf{x}), Y)\right] = \mu\mathbb{E}_{p^+}[\ell(g(\mathbf{x}), 1)] + (1-\mu)\mathbb{E}_{p^-}[\ell(g(\mathbf{x}), -1)]$$
$$= \mu\mathbb{E}_{p^+}[\ell(g(\mathbf{x}), 1)] - \mu\mathbb{E}_{p^+}[\ell(g(\mathbf{x}), -1)] + \mathbb{E}_{p^u}[\ell(g(\mathbf{x}), -1)] \qquad (5)$$

Here, $\ell$ denotes a loss function and $g$ represents the model.

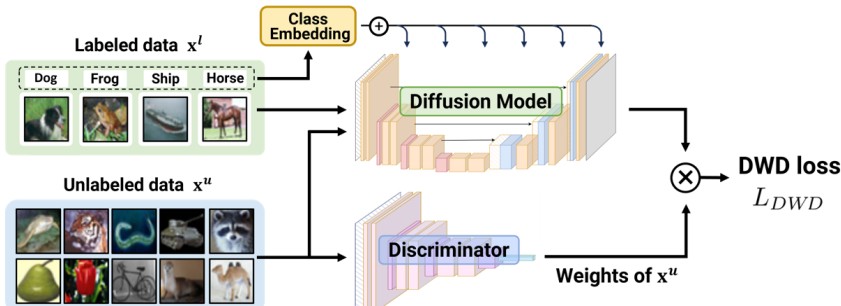

Figure 2: Schematic diagram of Discriminator-Weighted Diffusion (DWD). The conditional diffusion model is trained using both labeled and unlabeled data. The unlabeled data is utilized for unconditional training without class conditions. The pre-trained discriminator assigns weights to each unlabeled data sample to mitigate the potential negative impact of OOD samples.

## 4 Methodology

In this section, we introduce our data augmentation method, which leverages a diffusion model to generate synthetic data to address the scarcity of labeled data. As previously discussed, we start with training the diffusion model on labeled and unlabeled data while mitigating the class distribution mismatch. Subsequently, we employ the reverse diffusion process to transform unlabeled samples into synthesized labeled samples.

### 4.1 Training Diffusion Model

**A simple training scheme for SSL data**    The diffusion model trained only on the labeled data will inevitably overfit and merely generate replications due to their limited amount. It is thus essential to train the diffusion model on both labeled and unlabeled data, while taking advantage of the label information. We adopt a class-conditional diffusion model, shown in Figure 2, where labeled data are used for conditional training while unlabeled data are used for unconditional training, trained with the loss

$$L_{\text{semi}} = \mathbb{E}_{(\mathbf{x}_0,\mathbf{c}) \sim \mathcal{D}_l, t, \boldsymbol{\epsilon}}[||\boldsymbol{\epsilon}_\theta(\mathbf{x}_t, \mathbf{c}, t) - \boldsymbol{\epsilon}||_2^2] + \alpha \cdot \mathbb{E}_{\mathbf{x}_0 \sim \mathcal{D}_u, t, \boldsymbol{\epsilon}}[||\boldsymbol{\epsilon}_\theta(\mathbf{x}_t, t) - \boldsymbol{\epsilon}||_2^2], \qquad (6)$$

where $\alpha$ serves as a hyper-parameter that controls balance between labeled and unlabeled data, $\mathcal{D}_l$ and $\mathcal{D}_u$ represent labeled and unlabeled datasets. Intuitively, this straightforward objective aims to train the diffusion model to reconstruct not only the limited labeled data but also the abundant unlabeled data, thereby regularizing the model against overfitting to the labeled data. It is noteworthy that this approach shares similarities with the consistency regularization technique in the sense that the unlabeled data are utilized for regularization purposes.

**Discriminator**    However, when we train the diffusion model with the loss in (6), OOD unlabeled samples can have a negative impact on capturing important characteristics of the labeled data distribution. These samples should be made contribute less towards the overall training loss. In line with aforementioned filtering methods (Chen et al., 2020; Guo et al., 2020), we leverage a discriminator to weigh the unlabeled data instances. The discriminator is tasked with differentiating between positive samples that are closely aligned with the distribution of labeled data and negative samples that are irrelevant. To train the discriminator, we adopt PU learning with the training loss

$$L_{\text{disc}} = \mu \cdot \mathbb{E}_{\mathbf{x} \sim \mathcal{D}_l}[-\log \mathbf{d}_\phi(\mathbf{x}) + \log(1 - \mathbf{d}_\phi(\mathbf{x}))] + \mathbb{E}_{\mathbf{x} \sim \mathcal{D}_u}[-\log(1 - \mathbf{d}_\phi(\mathbf{x}))], \qquad (7)$$

where $\mathbf{d}_\phi$ denotes the discriminator parameterized by $\phi$, and $\mu$ represents the ratio of positive samples among unlabeled samples, i.e. belonging to one of the classes in the labeled dataset. This ratio can either be estimated from the dataset (Menon et al., 2015; Jain et al., 2016; Christoffel et al., 2016) or treated as a hyperparameter.

We then use the discriminator to assign weights on the unlabeled instances so that they align with the distribution of labeled data. A simple algebraic manipulation tells us that the following weight formula yields an unbiased loss estimation via importance sampling:

**Algorithm 1** Discriminator-Weighted Diffusion (DWD) - Training

---

**Input** Labeled dataset $\mathcal{D}_l$ and unlabeled dataset $\mathcal{D}_u$.
**Parameter** Learning rate $\eta_\theta$, $\eta_\phi$, hyper-parameter $\mu$, $\alpha$, and total number of iterations $K_\theta$, $K_\phi$.
Either pretrain the diffusion model $\epsilon_\theta$ on $\mathcal{D}_u$ or acquire an off-the-shelf pretrained diffusion model.
\# Train the discriminator $\mathbf{d}_\phi$ using objective (7):
**for** $n = 0, 1, 2, ..., K_\phi$ **do**
    Sample a batch of data $\mathbf{x}^l \sim \mathcal{D}_l$ and $\mathbf{x}^u \sim \mathcal{D}_u$.
    $\phi \leftarrow \phi - \eta_\phi \nabla_\phi \left[ \mu \cdot \mathbb{E}_{\mathbf{x}^l}[-\log \mathbf{d}_\phi(\mathbf{x}^l) + \log(1 - \mathbf{d}_\phi(\mathbf{x}^l))] + \mathbb{E}_{\mathbf{x}^u}[-\log(1 - \mathbf{d}_\phi(\mathbf{x}^u))] \right]$
**end for**
\# Finetune the diffusion model $\epsilon_\theta$ using the discriminator $\mathbf{d}_\phi$ and weighting function $w$ from (8):
**for** $n = 0, 1, 2, ..., K_\theta$ **do**
    Sample a batch of data $(\mathbf{x}^l, \mathbf{c}) \sim \mathcal{D}_l$ and $\mathbf{x}^u \sim \mathcal{D}_u$.
    $\theta \leftarrow \theta - \eta_\theta \nabla_\theta \left[ \mathbb{E}_{\mathbf{x}^l, t, \epsilon}[||\epsilon_\theta(\mathbf{x}_t^l, \mathbf{c}, t) - \epsilon||_2^2] + \alpha \cdot \mathbb{E}_{\mathbf{x}^u, t, \epsilon}[w(\mathbf{x}^u) \cdot ||\epsilon_\theta(\mathbf{x}_t^u, t) - \epsilon||_2^2] \right]$
**end for**
**Output** Learned diffusion model $\epsilon_\theta$ and learned discriminator $\mathbf{d}_\phi$.

---

**Proposition 4.1.** *Given an optimal discriminator $\mathbf{d}^*$, using the following importance weight on unlabeled data yields an unbiased estimation of the loss function with respect to the labeled data distribution:*

$$w(\mathbf{x}) = \frac{\mathbf{d}(\mathbf{x})}{\mu \, \mathbf{d}(\mathbf{x}) + (1-\mu)(1 - \mathbf{d}(\mathbf{x}))} \tag{8}$$

For a detailed proof, please refer to Appendix A. The final training loss for the diffusion model then becomes:

$$L_{\text{DWD}} = \mathbb{E}_{(\mathbf{x}_0, \mathbf{c}) \sim \mathcal{D}_l, t, \epsilon}[||\epsilon_\theta(\mathbf{x}_t, \mathbf{c}, t) - \epsilon||_2^2] + \alpha \cdot \mathbb{E}_{\mathbf{x}_0 \sim \mathcal{D}_u, t, \epsilon}[w(\mathbf{x}_0)||\epsilon_\theta(\mathbf{x}_t, t) - \epsilon||_2^2] \tag{9}$$

The overall training scheme, referred to as Discriminator-Weighted Diffusion (DWD), is outlined in Algorithm 1.

### 4.2 Seeding Data Generation with Unlabeled Instances

After training the diffusion model, a straightforward approach to generating synthetic data would be to start the reverse diffusion process from Gaussian noise. In our methodology, however, we begin the reverse process using a partially noised image of an unlabeled instance. As we will show in Section 5.3, we found that this approach leads to additional performance gains. It successfully transforms an OOD instance into an in-distribution sample while preserving some important characteristics in the original sample (see Figure 1). Thus we can exploit the diversity present in the unlabeled data when generating synthetic in-distribution data. The procedure is detailed in Algorithm 2 in the Appendix B.

We can think of two usage scenarios for the samples generated from the diffusion model: we could take the samples as pseudo-labeled synthetic data to enrich the labeled data and employ a fully supervised learning method, or take the samples as a transformed unlabeled data by discarding the class conditions and employ an SSL method. In the latter case, we expect that the SSL method will perform better, since the class distribution mismatch has been mitigated.

## 5 Experiments

To assess the effectiveness of DWD, we conduct experiments across a broad set of tasks in two settings. (1) **DWD-SL**: a fully supervised learning setting where the unlabeled dataset is converted to a pseudo-labeled dataset by replacing the instances with their transformations along with their class conditions as target labels; and (2) **DWD-UT**: an SSL setting where the unlabeled dataset is replaced by the transformed unlabeled samples and employ the baseline SSL method.

**Tasks** The SixAnimal task Oliver et al. (2018) uses CIFAR-10 dataset to classify six animal classes (bird, cat, deer, dog, frog, and horse). Following the setup in Huang et al. (2023), we sampled 400 images per class for the labeled dataset and included up to 4100 images per class in the unlabeled dataset. To investigate the impact of class distribution mismatch, we varied the mismatch percentage

Table 1: Performance comparison on four tasks. We report the mean accuracy averaged over three seeds, along with standard error. Top scores for each task are highlighted.

| Task Name | MixMatch | FixMatch | MPL | OpenMatch | Fix-A-Step | IOMatch | DWD-SL |
|---|---|---|---|---|---|---|---|
| SixAnimal ($\zeta = 75\%$) | $80.77\pm0.11$ | $82.50\pm0.16$ | $65.62\pm0.47$ | $80.34\pm0.21$ | $85.34\pm0.17$ | $83.05\pm0.16$ | $\mathbf{85.86\pm0.28}$ |
| CIFAR-10/100 | $71.02\pm0.32$ | $78.91\pm0.15$ | $70.95\pm0.34$ | $70.15\pm0.30$ | $74.60\pm0.31$ | $77.66\pm0.22$ | $\mathbf{80.05\pm0.14}$ |
| ImageNet-30 | $68.67\pm0.37$ | $70.07\pm0.26$ | $72.65\pm0.70$ | $72.78\pm0.48$ | $79.67\pm0.81$ | $79.23\pm0.29$ | $\mathbf{82.20\pm0.38}$ |
| ImageNet-100 | $69.30\pm0.41$ | $65.11\pm0.32$ | $68.43\pm0.33$ | $65.42\pm0.36$ | $65.80\pm0.49$ | $66.85\pm0.19$ | $\mathbf{82.81\pm0.31}$ |

Table 2: Performance of standard SSL methods before and after applying DWD-UT. Highlighted scores show significant increases without overlapping intervals.

| Task Name | MixMatch | Mixmatch +DWD-UT | FixMatch | FixMatch +DWD-UT | MPL | MPL +DWD-UT |
|---|---|---|---|---|---|---|
| SixAnimal ($\zeta = 75\%$) | $80.77\pm0.11$ | $\mathbf{84.72\pm0.22}$ | $82.50\pm0.16$ | $\mathbf{87.17\pm0.19}$ | $65.62\pm0.47$ | $\mathbf{83.88\pm0.18}$ |
| Cifar-10/100 | $71.02\pm0.32$ | $\mathbf{80.47\pm0.49}$ | $78.91\pm0.15$ | $\mathbf{83.80\pm0.25}$ | $70.95\pm0.34$ | $\mathbf{80.24\pm0.56}$ |
| ImageNet-30 | $68.67\pm0.37$ | $\mathbf{85.20\pm0.10}$ | $70.07\pm0.26$ | $\mathbf{81.87\pm0.61}$ | $72.65\pm0.70$ | $\mathbf{90.20\pm0.23}$ |
| ImageNet-100 | $69.30\pm0.41$ | $\mathbf{81.62\pm0.36}$ | $65.11\pm0.32$ | $\mathbf{80.38\pm0.34}$ | $68.43\pm0.33$ | $\mathbf{75.66\pm0.26}$ |

Table 3: Performance of open-set SSL methods before and after applying DWD-UT.

| Task Name | OpenMatch | OpenMatch +DWD-UT | Fix-A-Step | Fix-A-Step +DWD-UT | IOMatch | IOMatch +DWD-UT |
|---|---|---|---|---|---|---|
| SixAnimal ($\zeta = 75\%$) | $80.34\pm0.21$ | $\mathbf{85.71\pm0.33}$ | $85.34\pm0.17$ | $\mathbf{86.68\pm0.23}$ | $83.05\pm0.16$ | $\mathbf{87.20\pm0.13}$ |
| Cifar-10/100 | $70.15\pm0.30$ | $\mathbf{80.99\pm0.03}$ | $74.60\pm0.31$ | $\mathbf{79.02\pm0.75}$ | $77.66\pm0.22$ | $\mathbf{83.22\pm0.16}$ |
| ImageNet-30 | $72.78\pm0.48$ | $\mathbf{75.28\pm0.68}$ | $79.67\pm0.81$ | $\mathbf{82.95\pm0.45}$ | $79.23\pm0.29$ | $\mathbf{81.96\pm0.26}$ |
| ImageNet-100 | $65.42\pm0.36$ | $\mathbf{80.02\pm0.45}$ | $65.80\pm0.49$ | $\mathbf{76.23\pm0.37}$ | $66.85\pm0.19$ | $\mathbf{80.19\pm0.13}$ |

$\zeta$ in the composition of the unlabeled dataset. For example, when $\zeta = 75\%$, the unlabeled dataset contains three non-animal classes and one animal classes. We refer Appendix C for further details on the composition of the unlabeled dataset.

The CIFAR-10/100 task uses CIFAR-10 as the labeled dataset, and CIFAR-100 as the unlabeled dataset. While the whole CIFAR-100 was taken as the unlabeled dataset, we sampled 100 images per class from CIFAR-10 to simulate the scarce labeled data scenario. Notably, class labels in CIFAR-10 and CIFAR-100 do not exactly overlap, though there are similarities (e.g., "horse" in CIFAR-10 and "cattle" in CIFAR-100). Thus, this task complements the SixAnimal task, which had an exact class overlap between labeled and unlabeled data.

The ImageNet-30 task uses the ImageNet-30 dataset Hendrycks et al. (2019), which is a subset of ImageNet limited to 30 classes. Following Saito et al. (2021), we selected 5% of the data from the first 20 classes (approximately 50 samples per class) based on the alphabetical ordering of class names for the labeled dataset, and used the remaining data as the unlabeled dataset.

The ImageNet-100 task uses ImageNet-100 dataset, which sub-sampled 100 classes from ImageNet, as described in Cao et al. (2022). We divided these classes equally into 50% ID and 50% OOD classes following alphabetical order. From each ID class, we selected a small portion (10%) as labeled data with the remaining data forming the unlabeled dataset. This task assesses the effectiveness on higher-resolution images with a greater diversity of classes. Please refer to Appendix D for extensive results under various sizes of labeled dataset.

**Baseline SSL methods**  Since we use DWD to transform the unlabeled dataset into a dataset devoid of class distribution mismatch, we comprehensively consider as baseline SSL methods those which operate under the standard setting as well as the open-set setting. The baseline SSL methods under the standard setting are MixMatch (Berthelot et al., 2019), FixMatch (Sohn et al., 2020), and Meta Pseudo Labels (MPL) (Pham et al., 2021), and the methods under the open-set setting are OpenMatch (Saito et al., 2021), Fix-A-Step (Huang et al., 2023), and IOMatch (Li et al., 2023).

## 5.1 DWD-SL: Labeled Dataset Augmentation

Table 1 reports the comparative performance of DWD-SL against various baseline SSL methods. The results clearly demonstrate that DWD-SL substantially surpasses the performance of methods for standard SSL. This suggests that DWD successfully captures both the inherent distribution of the labeled data and the diversity of the unlabeled data, yielding highly effective synthetic labeled data for training.

Notably, DWD-SL even achieves competitive or superior results relative to open-set SSL methods. This advantage stems from DWD-SL's ability to transform the diversity of unlabeled data into labeled samples, rather than using this diversity merely as a form of regularization, as seen in baseline SSL methods. For further implementation details, please refer to Appendix C.

## 5.2 DWD-UT: Unlabeled Dataset Transformation

Table 2 and Table 3 show the impact of DWD-UT on the performances of the baseline SSL methods. From the results, we can confirm that DWD-UT effectively addresses the performance degradation caused by the class distribution mismatch. Notably, the most significant improvement was observed in the MPL method. Given that MPL is based on pseudo-labeling, this result indicates that DWD-UT effectively mitigates the confirmation bias associated with pseudo-labeling of OOD instances in the unlabeled dataset.

Figure 3 shows how performance degrades over the range of $\zeta$ in the SixAnimal task. We can clearly see that using DWD-UT makes SSL methods generally robust to the degree of class distribution mismatch, even though they operate under the assumption that there are no OOD instances in the unlabeled data.

We also remark that DWD-UT further improves the performance of open-set SSL methods. This implies that DWD-UT is orthogonal to the OOD mitigation mechanisms used in open-set SSL methods, offering a distinct contribution in addressing the class distribution mismatch: while most of the open-set SSL methods operate under the detect-and-filter paradigm to focus on excluding the OOD instances due to their negative impact, the diffusion model provides a powerful tool for making up the diversity lost by such exclusion.

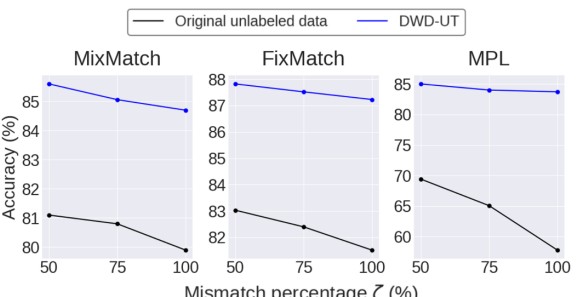

Figure 3: Standard SSL performance with varying $\zeta$.

It is also notable that DWD-UT frequently outperforms DWD-SL, indicating a synergistic effect between DWD-UT and baseline SSL methods. This may be attributed the underlying data selection mechanism in the SSL methods (e.g. thresholding used in pseudo-labeling, weighting function in filtering-based methods), which also contribute to selectively strengthen the impact of synthetic data.

Additionally, to provide direct evidence of DWD-UT's effectiveness in reducing the class distribution mismatch, we compute the minimum distance between each unlabeled data sample and the class centroids of labeled data in the latent space. We refer Appendix E for the results.

## 5.3 Ablation Studies

We carried out a series of ablation studies on DWD-UT, assessing different training schemes for the diffusion model and varying noise levels for perturbing seed images. For these studies, we employed MPL as the baseline SSL method.

Table 4 shows the results of ablation studies conducted on different training schemes for the diffusion model. Several observations can be drawn from these results. Firstly, the utilization of the discriminator to filter or reduce the weight of OOD instances culminates in a rather marginal performance improvement. Secondly, the incorporation of the diffusion model to generate synthetic data contributes to a substantial performance surge above the baseline, even when fine-tuned solely

Table 4: MPL performance using different training schemes. The notation $\epsilon_\theta[X]$ indicates the inclusion of component $X$ in finetuning the diffusion model. MPL + $\mathbf{d}_\phi$ represents that the discriminator is utilized for filtering unlabeled data.

| Training Method | SixAnimal | Cifar-10/100 | ImageNet-30 | ImageNet-100 |
|---|---|---|---|---|
| MPL | 65.62 | 70.95 | 72.65 | 68.43 |
| MPL + $\mathbf{d}_\phi$ | 67.19 | 71.73 | 83.60 | 70.74 |
| MPL + $\epsilon_\theta[\mathcal{D}_l]$ | 78.70 | 75.33 | 87.44 | 73.38 |
| MPL + $\epsilon_\theta[\mathcal{D}_l, \mathcal{D}_u]$ | 80.78 | 76.79 | 88.09 | 74.12 |
| MPL + $\epsilon_\theta[\mathcal{D}_l, \mathcal{D}_u, \mathbf{d}_\phi]$ | **83.88** | **80.24** | **90.20** | **75.66** |

Table 5: MPL performance on SixAnimal with varying noise levels. DWD-UT is not applied at $t = 0$.

| Noise Level | $t = 0$ | $t = 200$ | $t = 400$ | $t = 600$ | $t = 800$ | $t = 1000$ |
|---|---|---|---|---|---|---|
| Accuracy (%) | 65.62±0.47 | 73.65±0.28 | 82.03±0.18 | **83.88±0.18** | **83.83±0.11** | 82.06±0.13 |

Table 6: Performance of standard SSL and generative augmentation methods on ImageNet-30.

| Method | MixMatch | FixMatch | MPL | DPT | DA-Fusion | DWD-SL |
|---|---|---|---|---|---|---|
| Accuracy (%) | 68.67±0.37 | 70.07±0.26 | 72.65±0.70 | 78.28±0.48 | 75.26±0.39 | **82.20±0.28** |

with the labeled data. This suggests that transforming irrelevant unlabeled data is more effective than simply filtering them out. Lastly, the extra step of fine-tuning the diffusion model with the unlabeled dataset and applying discriminative weighting also offers a nontrivial advantage.

Table 5 shows the variations in performance with different noise levels during the data transformation process. The results indicate that introducing a moderate level of noise ($t = 600 \sim 800$) to the unlabeled data during the forward diffusion process, as opposed to initiating from pure Gaussian noise ($t = 1000$), enhances performance. Therefore, it can be inferred from these findings that the efficacy of DWD-UT is contingent upon the balance between the level of noise and the information contained in the unperturbed data. We remark that the optimal noise level can differ among data instances. While we fixed the noise level to $t = 600$ in our experiments for simplicity, determining the noise level individually for each sample presents a potential avenue for future research.

### 5.4 Comparison with Recent Diffusion-based Augmentation Approaches

We conducted further experiments on the ImageNet-30 task to compare the performance of DWD with those of DPT and DA-Fusion. To ensure fairness in comparison, we equalized the implementation of the model structure, data generation process, and the number of augmented data. As shown in Table 6, both DPT and DA-Fusion have demonstrated effectiveness, yet their performance falls short compared to that of DWD. This is because DPT, assuming no distribution shift, sometime generates wrongly labeled synthetic images due to the confirmation bias in pseudo labeld of OOD unlabeled images, and DA-Fusion only augments given labeled samples with subtle visual details (Please refer to Appendix I for examples). In contrast, DWD synthesizes new labeled samples by transforming diverse unlabeled data, successfully resolving the distribution mismatch.

### 5.5 Additional Experiments

We also conducted additional experiments to analyze computational costs, investigate the effect of the number of generated data, and assess hyper-parameter sensitivity. For detailed experimental results and analysis, please refer to Appendix F.

# 6    Conclusion

In this paper, we highlighted the potential of diffusion models for addressing class distribution mismatch in SSL. We introduced Discriminator-Weighted Diffusion (DWD), a semi-supervised training scheme that leverages a discriminator to identify OOD instances within the unlabeled data, facilitating effective training of the diffusion model. Our qualitative and quantitative results demonstrate that DWD captures both the characteristics of labeled data and the diversity of unlabeled data.

Notably, DWD exhibits a unique capability to convert irrelevant samples into relevant ones, making it compatible with other SSL methods and illustrating the orthogonality of our approach. Our extensive experiments show that DWD significantly enhances SSL performance in scenarios with class distribution mismatch. We hope that DWD will inspire future research focused on addressing distribution mismatch from a data-centric perspective.

## Acknowledgments and Disclosure of Funding

This work was supported by IITP grant funded by MSIT of Korea (No. RS-2020-II200940, No. RS-2022-II220311, No. RS-2019-II190075, No. RS-2024-00397310, No. RS-2024-00343989, No. RS-2024-00457882), and KAIST-NAVER Hypercreative AI Center.

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

# A Derivation for Proposition 4.1

**Problem setting.** We consider the two-sample problem setting of PU learning (Ward et al., 2009; Niu et al., 2016). The discriminator $\mathbf{d}(\mathbf{x})$ aims to solve a binary classification problem to classify input data $\mathbf{x}$ into negative or positive class $y$. Let $X \in \mathbb{R}^d$ and $Y \in \{-1, +1\}$ denote the input and output random variables, $p(\mathbf{x}, y)$ be the joint probability density function of $(X, Y)$, and $p^+(\mathbf{x}) = p(\mathbf{x} \,|\, Y = +1)$, $p^-(\mathbf{x}) = p(\mathbf{x} \,|\, Y = -1)$ be the positive and negative conditional probability density functions respectively. The labeled and unlabeled data are assume to be sampled from $p^+(\mathbf{x})$ and $p(\mathbf{x}) = \mu \, p^+(\mathbf{x}) + (1 - \mu) \, p^-(\mathbf{x})$ respectively, where $\mu = p(Y = +1)$ is class prior.

*Proof.* The goal of the discriminator can be formulated as maximizing following objective

$$
\begin{aligned}
J &= \mathbb{E}_{\mathbf{x} \sim p^+(\mathbf{x})} \left[ \log \left( \mathbf{d}\left( \mathbf{x} \right) \right) \right] + \mathbb{E}_{\mathbf{x} \sim p^-(\mathbf{x})} \left[ \log \left( 1 - \mathbf{d}\left( \mathbf{x} \right) \right) \right] \\
&= \int p^+(\mathbf{x}) \log \left( \mathbf{d}\left( \mathbf{x} \right) \right) + p^-(\mathbf{x}) \log \left( 1 - \mathbf{d}\left( \mathbf{x} \right) \right) \mathrm{d}\mathbf{x}
\end{aligned}
\tag{10}
$$

First order optimality condition gives

$$
\frac{p^+(\mathbf{x})}{\mathbf{d}(\mathbf{x})} - \frac{p^-(\mathbf{x})}{1 - \mathbf{d}(\mathbf{x})} = 0
\tag{11}
$$

Rearranging (11), we can easily show that the optimal discriminator $\mathbf{d}^*(\mathbf{x})$ satisfies

$$
\mathbf{d}^*(\mathbf{x}) = \frac{p^+(\mathbf{x})}{p^+(\mathbf{x}) + p^-(\mathbf{x})}
\tag{12}
$$

Substituting (12) to (8), we have

$$
w(\mathbf{x}) = \frac{\mathbf{d}^*(\mathbf{x})}{\mu \, \mathbf{d}^*(\mathbf{x}) + (1 - \mu)(1 - \mathbf{d}^*(\mathbf{x}))} = \frac{p^+(\mathbf{x})}{\mu \, p^+(\mathbf{x}) + (1 - \mu) \, p^-(\mathbf{x})} = \frac{p^+(\mathbf{x})}{p(\mathbf{x})}
\tag{13}
$$

Finally, we can use $w(\mathbf{x})$ as importance weights because

$$
\mathbb{E}_{\mathbf{x}_0 \sim p(\mathbf{x})}[w(\mathbf{x}_0)||\boldsymbol{\epsilon}_\theta(\mathbf{x}_t, t) - \boldsymbol{\epsilon}||_2^2] = \int \frac{p^+(\mathbf{x}_0)}{p(\mathbf{x}_0)} p(\mathbf{x}_0) ||\boldsymbol{\epsilon}_\theta(\mathbf{x}_t, t) - \boldsymbol{\epsilon}||_2^2 \mathrm{d}\mathbf{x} = \mathbb{E}_{\mathbf{x}_0 \sim p^+(\mathbf{x})}[||\boldsymbol{\epsilon}_\theta(\mathbf{x}_t, t) - \boldsymbol{\epsilon}||_2^2]
\tag{14}
$$

# B Data Generation Process

The data generation process is detailed in Algorithm 2. In our experiments, we generate one image per unlabeled image. For the impact of the number of generated data, please refer to Appendix F. We chose to generate images before the classification task, rather than during each batch iteration of training the classification models. Additionally, while we described the sampling process in DDPM style, DPM-Solver[2](Lu et al., 2022) is utilized in implementation for computational efficiency.

---

**Algorithm 2** DWD - Image-Seeded Generation

---

**Input** Unlabeled data $\mathbf{x}^u$ and diffusion model $\boldsymbol{\epsilon}_\theta$.
**Parameter** Time step $t$.
Sample Gaussian noise: $\boldsymbol{\epsilon} \sim \mathcal{N}(\mathbf{0}, \mathbf{I})$
Forward diffusion process to time step $t$: $\mathbf{x}_t^u = \sqrt{\bar{\alpha}_t} \mathbf{x}^u + \sqrt{(1 - \bar{\alpha}_t)} \boldsymbol{\epsilon}$
Randomly select class condition $\mathbf{c}$.
\# Reverse diffusion process starting at $\mathbf{x}_t^u$:
**for** $i = t, t - 1, ..., 1$ **do**
$\quad \mathbf{z} \sim \mathcal{N}(\mathbf{0}, \mathbf{I})$ if $i > 1$ else $\mathbf{z} = 0$
$\quad \mathbf{x}_{i-1}^u = \frac{1}{\sqrt{\alpha_i}} \left( \mathbf{x}_i^u - \frac{1 - \alpha_t}{\sqrt{1 - \bar{\alpha}_i}} \boldsymbol{\epsilon}_\theta(\mathbf{x}_i^u, \mathbf{c}, i) \right) + \sqrt{\beta_i} \mathbf{z}$
**end for**
**Output** Transformed data $\mathbf{x}_0^u$ and class condition $\mathbf{c}$

---

[2]https://github.com/LuChengTHU/dpm-solver, MIT License

# C Implementation Details

**Diffusion model and discriminator.** For all experiments, we use official implementation of latent diffusion model (Rombach et al., 2022), which is publicly available[3]. Since latent diffusion models perform diffusion processes in the embedded latent space, there is a trade-off between computational cost and generation quality depending on the downsampling factor $f$ Rombach et al. (2022). Therefore, we adjust $f$ based on dataset scale: $f = 2$ for small-scale datasets and $f = 8$ for large-scale datasets. The batch sizes for labeled and unlabeled data, denoted as $B_l$ and $B_u$ respectively, are set to $B_l = 16$, $B_u = 112$ for small-scale datasets, and $B_l = 4$, $B_u = 12$ for large-scale datasets. We follow the Rombach et al. (2022) for other configuration such as learning rate, optimizer, scheduler, etc. After the pre-training phase, we train the models for 200K iterations for Cifar-10, and 1M iterations for ImageNet. For the discriminator, we employ the ResNet-18 He et al. (2016) followed by 2 MLP layers. We train the discriminator using AdamW Loshchilov & Hutter (2019) optimizer with 0.0002 initial learning rate and 0.0001 weight decay. We treat $\mu$ as a hyper-parameter, and set it within {0.25, 0.33, 0.5}. Another hyper-parameter $\alpha$, which control the balance between labeled and unlabeled, we set 5 in all tasks.

**Common configuration for DWD-SL and DWD-UT.** To ensure fair evaluation, task-specific settings were established for both DWD-SL and DWD-UT. For tasks associated with Cifar-10, the Wide ResNet-28-2 architecture Zagoruyko & Komodakis (2016)) was employed, with training conducted using the AdamW optimizer at an initial learning rate of 0.03 across 256 epochs 1,024 iterations per epoch. In the ImageNet-30 task, we follow the settings from Saito et al.(2021) and Li et al.(2023). Specifically, we employed the ResNet-18 architecture He et al. (2016), and train for 100 epochs with 1,024 iterations per epoch using AdamW 0.1 initial learning rate.

**DWD-SL specific configuration.** In all DWD-SL tasks, we maintained a 1:1 ratio between labeled and unlabeled samples within each batch. More specifically, we set $B_l = B_u = 64$ for SixAnimal and Cifar-10/100, and $B_l = B_u = 32$ for ImageNet-30. We applied RandAugment Cubuk et al. (2020), widely used in the SSL field to achieve robust results, to both original labeled and DWD-SL data. Additionally, we applied label smoothing to the cross-entropy loss, following the approach used in MPL Pham et al. (2021). The starting time step $t$ for the reverse diffusion process was set to 600.

**DWD-UT specific configuration.** For SixAnimal and Cifar-10/100 tasks, we used $B_l = 64$ and $B_u = 448$, while for ImageNet-30, we used $B_l = 32$ and $B_u = 32$. Since ImageNet-30 includes out-of-distribution (OOD) classes in the test set while standard SSL methods inherently cannot identify OOD classes, we removed OOD samples from the test set for a fair comparison. Additionally, IOMatch evaluates the performance using balanced accuracy, which classifies all OOD classes as a additional single class. This could also be an unfair comparison. Therefore, we evaluated only on the closed set for all tasks. Except aforementioned, we follow their papers for method-specific hyper-parameters and setting.

**Unlabeled data composition in SixAnimal with varying $\zeta$.** We configure the SixAnimal task exactly following Chapelle et al. (2009), Huang et al. (2023). The Table 7 shows the composition of labeled / unlabeled set of SixAnimal task according to mismatch percentage ($\zeta$).

Table 7: Configuration of labeled/unlabeled class mismatch scenario in CIFAR-10 SixAnimal task. The bold text of unlabeled set represent the OOD classes.

|  | Labeled set | Unlabeled set |
|---|---|---|
| $\zeta = 25\%$ | Bird, Cat, Deer, Dog, Frog, Horse | **Airplane**, Dog, Frog, Horse |
| $\zeta = 50\%$ | Bird, Cat, Deer, Dog, Frog, Horse | **Airplane, Car**, Frog, Horse |
| $\zeta = 75\%$ | Bird, Cat, Deer, Dog, Frog, Horse | **Airplane, Car, Ship**, Horse |
| $\zeta = 100\%$ | Bird, Cat, Deer, Dog, Frog, Horse | **Airplane, Car, Ship, Truck** |

---

[3]https://github.com/CompVis/latent-diffusion, MIT License

# D  Extended Evaluation on Class Mismatch and Labeled Data Ratios

**Different mismatch ratios of ID and OOD classes in the SixAnimal task.**  As shown in Figure 3 of our paper, we conducted experiments on the SixAnimal task across various ratios of ID and OOD classes ($\zeta = 50\%$, $75\%$, $100\%$). To extend these findings, we included additional experiments with a lower mismatch ratio, $\zeta = 25\%$. Since DWD is designed to address class distribution mismatch, its effectiveness is expected to decrease as the mismatch ratio lowers. However, we observed that DWD was still able to improve the baseline method in the $\zeta = 25\%$ case, although the performance gain was relatively diminished.

Table 8: Performance evaluation on the SixAnimal task with various ratios of ID and OOD classes ($\zeta$).

| Mismatch ratio | MixMatch | MixMatch +DWD-UT | FixMatch | FixMatch +DWD-UT | MPL | MPL +DWD-UT |
|---|---|---|---|---|---|---|
| $\zeta = 25\%$ | 83.78 | **85.89 (+2.11)** | 84.56 | **85.81 (+1.25)** | 79.70 | **84.76 (+ 5.06)** |
| $\zeta = 50\%$ | 81.16 | **84.83 (+3.67)** | 83.23 | **87.78 (+4.55)** | 69.43 | **85.02 (+15.59)** |
| $\zeta = 75\%$ | 80.77 | **84.72 (+3.95)** | 82.50 | **87.17 (+4.67)** | 65.62 | **83.88 (+18.26)** |
| $\zeta = 100\%$ | 79.90 | **84.08 (+4.18)** | 81.51 | **87.03 (+5.52)** | 57.77 | **83.73 (+25.96)** |

**Varying the size of labeled data in the ImageNet-100 task.**  To broaden our evaluation, we conducted additional experiments on the ImageNet-100 dataset, varying the amount of labeled data. Specifically, we used either 10% or 30% of each ID class as labeled data, with the remaining samples forming the unlabeled set. As shown in Table 9, DWD remains effective with different amounts of labeled data, demonstrating strong performance at both 10% and 30% sampling ratios. Notably, the performance gain is more pronounced at the 10% sampling ratio, as the advantages of data augmentation become clearer with smaller datasets. However, DWD's effectiveness may be constrained when labeled data is extremely limited, as the diffusion model may struggle to accurately represent the labeled data distribution. This limitation is also observed in other generative augmentation methods.

Table 9: Performance evaluation on the ImageNet-100 task with varying sampling ratio ($\gamma$).

| Sampling ratio | MixMatch | FixMatch | MPL | OpenMatch | Fix-A-Step | IOMatch | DWD-SL |
|---|---|---|---|---|---|---|---|
| $\gamma = 10\%$ | 69.30 | 65.11 | 68.43 | 65.42 | 65.80 | 66.85 | **82.81** |
| $\gamma = 30\%$ | 77.88 | 75.83 | 71.67 | 77.31 | 73.90 | 76.02 | **84.43** |

| Sampling ratio | MixMatch | MixMatch +DWD-UT | FixMatch | FixMatch +DWD-UT | MPL | MPL +DWD-UT |
|---|---|---|---|---|---|---|
| $\gamma = 10\%$ | 69.30 | **81.62 (+12.32)** | 65.11 | **80.38 (+15.27)** | 68.43 | **75.66 (+7.23)** |
| $\gamma = 30\%$ | 77.88 | **82.26 (+ 4.38)** | 75.83 | **81.35 (+ 5.52)** | 71.67 | **77.49 (+5.82)** |

| Sampling ratio | OpenMatch | OpenMatch +DWD-UT | Fix-A-Step | Fix-A-Step +DWD-UT | IOMatch | IOMatch +DWD-UT |
|---|---|---|---|---|---|---|
| $\gamma = 10\%$ | 65.42 | **80.02 (+14.60)** | 65.80 | **76.23 (+10.43)** | 66.85 | **80.19 (+13.34)** |
| $\gamma = 30\%$ | 77.31 | **81.50 (+ 4.19)** | 73.90 | **78.43 (+ 4.53)** | 76.02 | **81.52 (+ 5.50)** |

# E Calculating Distance in Latent Space

We extract features (labeled, original unlabeled, transformed unlabeled) from four datasets using a pre-trained ResNet-50. The features were normalized, and the pair-wise Euclidean distances between each unlabeled sample and the nearest class centroid is visualized using Gaussian Kernel Density Estimation (KDE). As shown in Figure 4, the minimum distances successfully decrease after DWD's data transformation. This result indicates that using DWD to transform unlabeled data does more than just make it look similar to labeled data; it also makes the data semantically similar in the latent space, showing a deeper level of similarity beyond just appearance.

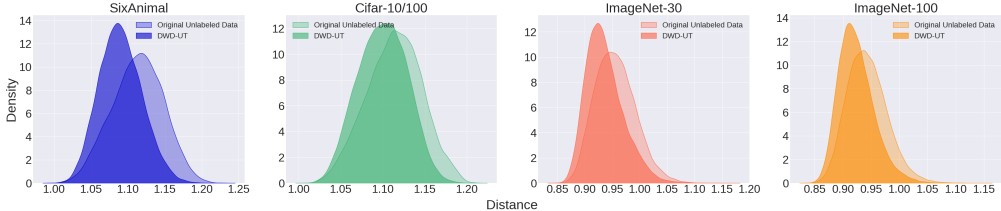

Figure 4: The distribution of the minimum distance between each unlabeled data sample and the class centroids of labeled data.

# F Additional Experiments

**Computational cost analysis.** The main limitation of our work is the additional computation required by incorporating the diffusion model. On the CIFAR-10/100 task, we measured the wall-clock times and memory consumption for each stage of DWD: pretraining, finetuning (including discriminator training), and sampling. We compare the computation costs with standard SSL methods, and the results are reported in Table 10. The additional computation for DWD is not overly burdensome compared to the baselines. It is worth noting that while we include pretraining costs for completeness, these can be omitted when off-the-shelf pre-trained diffusion models are available.

Table 10: Computational Cost Comparison.

|  | DWD | | | Baselines | | |
|---|---|---|---|---|---|---|
|  | Pretraining | Finetuning | Sampling | Mixmatch | Fixmatch | MPL |
| Elapsed time (Hours) | 13.8 | 9.7 | 0.1 | 9.2 | 7.4 | 6.7 |
| Memory (GB) | 7.1 | 8.0 | 7.0 | 4.9 | 5.5 | 7.4 |

[Machine specification] GPU : NVIDIA GeForce RTX 3090 Ti, CPU : Intel(R) Core(TM) i9-10980XE

**Effect of the number of generated data.** We conducted additional experiments using DWD-SL on the Cifar-10/100 task, varying the number of generated data denoted as N. The results are summarized in the Table 11. Here, N = 60K corresponds to the original setting in the paper where one synthetic labeled sample per one unlabeled sample is generated. We observed further improvement at N = 120K, with a slight deterioration thereafter. Note that a similar trend was previously reported in Figure 7(c) of Appendix G in You et al. (2023). A reasonable explanation for the performance deterioration is that an excessive value of N could cause the classifier to be dominated by synthetic data, thereby neglecting real data, as suggested by You et al. (2023).

Table 11: Effect of the number of generated data

| N | 6K | 30K | 60K | 120K | 240K |
|---|---|---|---|---|---|
| Accuracy (%) | 77.64 | 79.17 | 80.05 | **81.24** | 81.07 |

**Hyper-parameters sensitivity.** We conducted additional experiments on the SixAnimal task ($\zeta$=75%) using DWD-SL to assess DWD's sensitivity to the hyper-parameters. Again, $\alpha$ serves as weight to control the balance between labeled and unlabeled data (Eq. 9) and $\mu$ is treated as positive sample ratio of unlabeled data to train discriminator (Eq. 7) in our training scheme. We observed that a wide range of $\alpha$ and $\mu$ values successfully outperform most of the baselines (refer to Table 1 in our paper). Regarding $\alpha$, an extremely small value may cause the diffusion model training to focus excessively on the labeled data, failing to reflect the diversity of the unlabeled data and potentially leading to overfitting. Conversely, an extremely large value may cause the training to skew towards the unlabeled data, failing to properly capture the labeled data distribution. In our experiments, an $\alpha$ value around 3 achieves an appropriate trade-off. Regarding $\mu$, the optimal value is near the true ratio 1 - $\zeta$, as expected.

Table 12: Performance of DWD with various $\alpha$.

| $\alpha$ | 1 | **3** | 5 | 10 |
|---|---|---|---|---|
| Accuracy (%) | 84.01 | **85.86** | 83.83 | 83.51 |

Table 13: Performance of DWD with various $\mu$.

| $\mu$ | 0.125 | **0.25** | 0.33 | 0.55 |
|---|---|---|---|---|
| Accuracy (%) | 84.56 | **85.86** | 85.33 | 84.72 |

## G Images Corresponding to Discriminator's Output

As shown in the Figure 5, our discriminator successfully identifies the ID/OOD samples. ImageNet-30 consist of 20 in-domain (ID) classes (acorn, airliner, ambulance, american alligator, banjo, barn, bikini, digital clock, dragonfly, dumbbell, forklift, goblet, grand piano, hotdog, hourglass, manhole cover, mosque, nail, parking meter, pillow) and 10 out-of-domain (OOD) classs (revolver, rotary dial telephone, schooner, snowmobile, soccer ball, stingray, strawberry, tank, toaster, volcano)

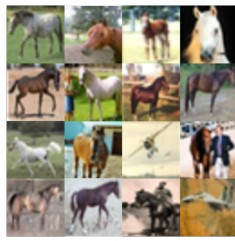 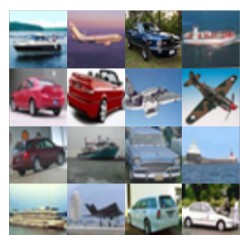 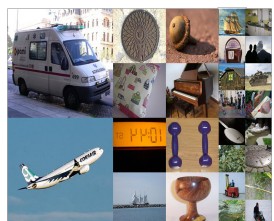 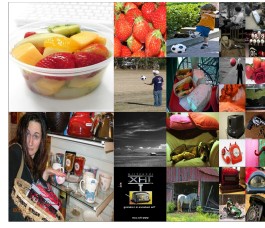

(a) High scored samples    (b) Low scored samples    (c) High scored samples    (d) Low scored samples

Figure 5: Selected unlabeled samples based on discriminator's output on the SixAnimal (a, b) and ImageNet-30 (c, d).

## H   Generated Images

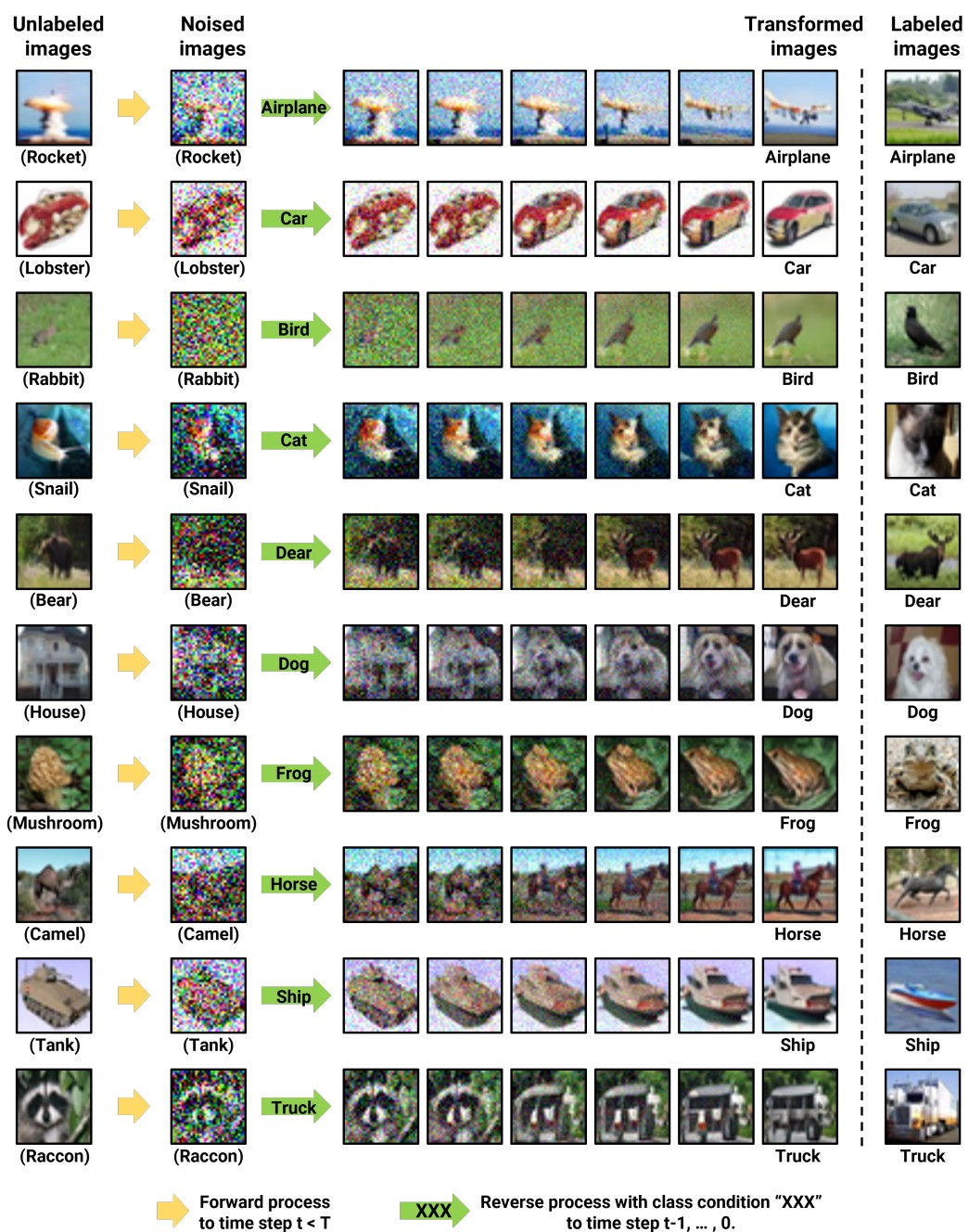

Figure 6: Selected Examples of Data Geneartion Process.

# I  Generated Images from DPT and DA-Fusion

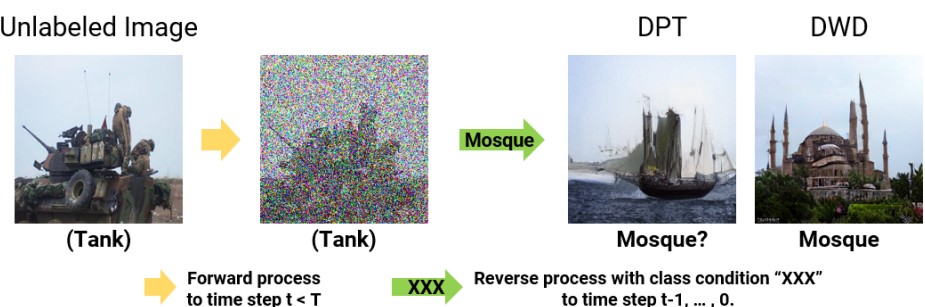

Figure 7: Generated images from DPT and DWD. DPT sometimes generates incorrectly labeled samples (e.g., an image of a schooner, which is an OOD class, labeled as a mosque). Note that while DPT originally samples images from scratch, we applied our data generation algorithm to DPT for comparison.

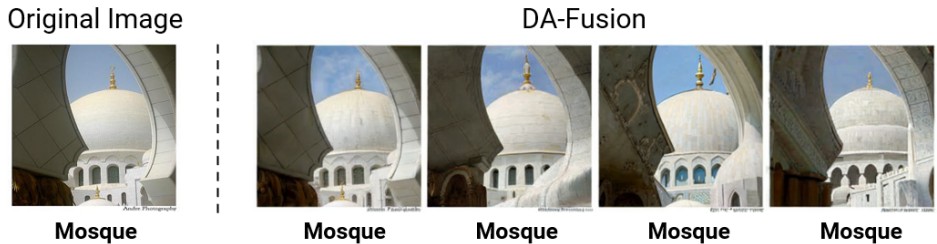

Figure 8: Generated images from DA-Fusion. DA-Fusion only augments given labeled images with subtle visual details.

