# OpenReview forum: "Data Augmentation with Diffusion for Open-Set Semi-Supervised Learning"
_NeurIPS.cc/2024/Conference — NeurIPS 2024 poster_

### Official Review · Reviewer_5EHv · 2024-07-05

**Soundness:** 3
**Presentation:** 2
**Contribution:** 3
**Rating:** 6
**Confidence:** 3

**Summary:**

This paper aims to address the challenge of utilizing unlabeled data in SSL, especially when there is a mismatch in class distributions between labeled and unlabeled data. The authors propose to leverage diffusion models to convert unlabeled data, especially out-of-distribution data, into in-distribution samples. Besides, a discriminator is jointly trained to filter out potentially irrelevant data. Experiments on image classification tasks show that the proposed method can greatly improve the performance of existing SSL methods, especially in scenarios with large class distribution mismatch.

**Strengths:**

1. The paper proposes a novel approach to tackle the class distribution mismatch problem in SSL by leveraging diffusion models to convert unlabeled or OOD samples to in-distribution samples. This is a creative combination of ideas from generative modeling and SSL and is well-motivated.

2. The results demonstrate clear improvements over state-of-the-art SSL methods, especially in challenging scenarios with large distribution mismatch. The proposed method can also be served as a plug-in approach to enhance existing SSL methods.

3. The paper provides good visualizations that help understand the generated samples and the effectiveness of the method and discriminator's scores.

**Weaknesses:**

1. The use of diffusion models introduces significant computational costs compared to standard SSL methods. While this is acknowledged, a more detailed analysis of the trade-offs between performance gains and computational cost would be valuable.

2. More comparisons with related generative augmentation in SSL are needed. The paper only compares with the two closely-related work, DPT and DA-Fusion, on one dataset (ImageNet-30). Comparisons to advanced data augmentation techniques used in SSL on more datasets would provide more context.

3. The experiments are limited to relatively two~three small-scale datasets (The SixAnimal dataset is sampled from CIFAR10). It's unclear how well the approach would scale to larger, more complex datasets.

4. The paper focuses primarily on successful cases. A discussion of scenarios where DWD might not work well or potential limitations would provide a more balanced view.

5. While the discriminator is a key component, the presentation of the discriminator and the related positive-unlabeled learning is a bit confusing. More explanation on how the discriminator is trained and why the PU learning is adopted would be helpful. Also, have you considered alternative designs for the discriminator?

6. The method introduces several hyper-parameters, such as the $\alpha$, $\mu$. A more thorough discussion or analysis of the sensitivity to these parameters would strengthen the paper.

**Questions:**

1. The computation cost for different generative augmentation methods can be varied. How do you ensure the comparisons are fair?
2. How does the number of labeled data affect the effectiveness of the proposed method?

**Limitations:**

Some limitations are discussed in Appendix.

---

> ### Author Rebuttal · Authors · 2024-08-07
>
> We appreciate your constructive feedback.
>
> **W1.** To investigate trade-offs between performance gains and computational cost, we conducted additional experiments on the CIFAR-10/100 tasks, varying U-Net depths $d$ (i.e., the number of residual blocks per downsample) in the diffusion model. The results are summarized in the table below:
>
> |                                    | $d = 1$  | $d = 2$ | $d = 4$   |
> |------------------------------------|----------|-------------------------------|-----------|
> | **FixMatch + DWD-UT Accuracy (%)**    | 80.7     | 83.8                          | 84.2      |
> | **Model parameters (M)**           | 50.0     | 70.1                          | 110.2     |
> | **Elapsed time (Hours)**           | 15.6     | 21.5                          | 32.3      |
>
> As expected, increasing the depth of the U-Net led to further performance improvements. Thus, by adjusting the depth according to available computational resources, one can effectively balance the trade-off between model performance and computational cost.
> Additionally, the architecture of diffusion models (e.g., U-ViT [A], DiT [B]) can also influence computational cost and performance, which we plan to explore in future work.
>
> **W2.** First of all, we would like to highlight that DPT and DA-fusion demonstrate state-of-the-art performance in generative data augmentation techniques. To the best of our knowledge, other existing generative approaches based on VAE [C] or GAN [D] are limited to low-resolution data, which is why we did not include them as baselines. Additionally, we remark that some of our baselines, such as Fixmatch, Mixmatch, and Fix-A-Step, can be also considered as data augmentation techniques. Therefore, our experimental results demonstrate that DWD outperforms the well-known data augmentation techniques used in SSL. If you could suggest a manageable set of other advanced augmentation methods in SSL, we would be more than happy to conduct further experiments during the author-reviewer discussion period.
>
> **W3, Q2.** To verify DWD’s scalability to larger datasets and investigate the impact of the size of labeled data on DWD, we conducted additional experiments on the ImageNet-100 dataset, varying the size of labeled data. To construct the ImageNet-100 dataset, we sub-sampled 100 classes from ImageNet, as described in [E]. We divided these classes equally into 50% ID and 50% OOD classes, following alphabetical order. From each ID class, we selected a small portion (10% or 30%) as labeled data, with the remaining data forming the unlabeled set. We report the results in Table B in the PDF.
>
> As shown in the table, DWD-SL outperforms the baselines, and DWD-UT significantly enhances baseline performance. From these results, we can conclude that DWD can be effectively applied to larger datasets.
> In addition, DWD proves effective with varying amounts of labeled data, performing well with both 10% and 30% sampling ratios. Notably, the performance gain is greater with the 10% sampling ratio compared to the 30% ratio, as the benefit of data augmentation is more pronounced when the dataset is smaller. However, it is important to note that DWD’s effectiveness is expected to be marginal when the amount of labeled data is extremely small, as the diffusion model struggles to accurately capture the labeled data distribution. We remark that this limitation is also common to other generative augmentation approaches.
>
> **W4.** We acknowledge the importance of discussing potential limitations to provide a more balanced view. To investigate scenarios where DWD might not perform well, we extended Figure 3 in our paper to include a lower degree of class distribution mismatch ($\zeta$=25%). We report the results in Table A in the PDF.
>
> Since DWD is designed to resolve class distribution mismatch, its effectiveness is expected to decrease with a lower mismatch ratio. However, we observed that DWD was still able to improve the baseline method in the $\zeta$ = 25% case, although the performance gain was relatively small.
>
> As previously discussed in response to [W3,Q2], another limitation of DWD is its limited applicability with extremely small size of labeled data. Although DWD can benefit from the diversity of unlabeled data, it must first learn the distribution of the labeled data.
>
> **W6.** We conducted additional experiments on the SixAnimal task ($\zeta$ = 75%) using DWD-SL to assess DWD's sensitivity to the hyper-parameters.
>
> 1)  **$\alpha$** : We varied  across values of {1, 3, 5, 10}, and the results are presented in the table below:
>
> |                | $\alpha = 1$   | $\alpha = 3$ | $\alpha = 5$  | $\alpha = 10$ |
> |----------------|----------------|--------------|---------------|---------------|
> | Accuracy (%)   | 84.0           | 85.9         | 83.8          | 83.5          |
>
> 2) **$\mu$** : We varied  across values of {0.125, 0.25, 0.33, 0.5}, and the results are presented in the table below:
>
> |                | $\mu = 0.125$   | $\mu = 0.25$ | $\mu = 0.33$ | $\mu = 0.5$ |
> |----------------|-----------------|--------------|--------------|-------------|
> | Accuracy (%)   | 84.5            | 85.9         | 85.3         | 84.7        |
>
> We observed that a wide range of $\alpha$ and $\mu$ values successfully outperform most of the baselines (refer to table 1 in our paper).
>
> Regarding $\alpha$, an extremely small value may cause the diffusion model training to focus excessively on the labeled data, failing to reflect the diversity of the unlabeled data and potentially leading to overfitting. Conversely, an extremely large value may cause the training to skew towards the unlabeled data, failing to properly capture the labeled data distribution. In our experiments, an $\alpha$ value around 3 achieves an appropriate trade-off.
>
> Regarding $\mu$, the optimal value is near the true ratio $1-\zeta$, as expected.

---

> ### Author Response · Authors · 2024-08-07
> **Response to [Q1]**
>
> As you pointed out, even when the types of diffusion models and training processes are standardized, the total computational cost can vary across different generative augmentation methods. This is because each method includes unique auxiliary processes (e.g., training a classifier with partially labeled data, and extracting features in DPT, training a discriminator in DWD) in addition to the common computational costs associated with training the diffusion model.
>
> To ensure that the additional processes introduced in DWD do not compromise the fairness of comparisons, we conducted supplementary experiments to measure the extra cost of DWD compared to the extra cost of DPT on the CIFAR-10/100 tasks. The results are as follows:
>
> |                          | Diffusion model training | Extra cost of DWD | Extra cost of DPT |
> |--------------------------|-------------|------------------------|----|
> | **Elapsed time (Hours)** | 21.8        | 1.7                    |1.1|
> | **Memory (GB)**          | 8.0         | 2.9                    |2.0|
>
> [Machine specification] GPU : NVIDIA GeForce RTX 3090 Ti, CPU : Intel(R) Core(TM) i9-10980XE
>
> The results showed that the increase in computational cost due to the additional processes in DWD was similar to that of DPT. Therefore, we can ensure that the fairness of the comparison is not compromised.
>
> **References**
>
> [A] Bao et al., “All are worth words: a vit backbone for score-based diffusion models”, NeurIPS, 2022. Workshop on Score-Based Methods.
>
> [B] Peebles et al., "Scalable diffusion models with transformers." ICCV, 2023.
>
> [C] Li et al., “Max-margin deep generative models for (semi-) supervised learning”, IEEE Transactions on Pattern Analysis and Machine Intelligence, 2017.
>
> [D] Li et al., “Triple generative adversarial networks,” IEEE Transactions on Pattern Analysis and Machine Intelligence, 2021.
>
> [E] K Cao et al., “OPEN-WORLD SEMI-SUPERVISED LEARNING.”, ICLR 2022.
>
> [F] Chen et al., “Semi-supervised learning under class distribution mismatch“, AAAI, 2020.
>
> [G] Huang et al., “They are not completely useless: Towards recycling transferable unlabeled data for class-mismatched semi-supervised learning”, IEEE Transactions on Multimedia, 2022.
>
> [H] You et al., “Diffusion Models and Semi-Supervised Learners Benefit Mutually with Few Labels”, NeurIPS, 2023.

---

> > ### Comment · Reviewer_5EHv · 2024-08-12
> >
> > I thank the authors for their detailed responses and efforts to conduct additional experiments. I am inclined to accept this paper.

---

> > > ### Author Response · Authors · 2024-08-14
> > > **Official Comment by Authors**
> > >
> > > Thank you very much for the score improvement and your constructive feedback. We will further polish the paper in the final revision. Thank you!

---

### Official Review · Reviewer_9H8V · 2024-07-11

**Soundness:** 2
**Presentation:** 3
**Contribution:** 3
**Rating:** 6
**Confidence:** 4

**Summary:**

The paper proposes an approach that leverages a diffusion model to enrich labeled data using both labeled and unlabeled samples to try to solve the traditional ssl method struggling in the real-world scenarios, i.e., a large number of irrelevant instances in the unlabeled data that do not belong to any class in the labeled data. Specifically, authors combine diffusion model training with a discriminator, and convert an irrelevant instance into a relevant one. Empirically, the data augmentation approach which is proposed by this paper significantly enhances the performance of SSL methods.

**Strengths:**

1. The motivation is clear, and the method is effective and easy to follow.
2. The combination of DWD-UT and semi-supervised methods is interesting, and the results are impressive.
3. The results in the appendix address many of my questions, and the analysis is comprehensive.

**Weaknesses:**

1. There is a lack of analysis regarding the structure and training methods of the diffusion model networks, such as DiT and DDPM. Is DiT also effective for DWD-UT and DWD-SL?
2. The ImageNet-30 dataset is small. It remains to be verified whether the method can perform well in an open semi-supervised setting.

**Questions:**

1. As mentioned in Weakness 2, can the method be applied effectively to larger datasets?
2. Can this method be applied to other generative model, such as GAN, Flow?
3. How about the generation performance? can author evaluate the common generation metric such as FID, IS?

**Limitations:**

yes

---

> ### Author Rebuttal · Authors · 2024-08-07
>
> We appreciate your constructive feedback.
>
> **W1.** Thank you for bringing up the different diffusion model network structures and training methods for further improving our work. Given that DPT [A], which is closest prior work to our methodology, has demonstrated a remarkable performance with U-ViT [B], we anticipate that DiT [C], which is also based on ViT, will perform well with DWD. However, due to current limitations in computational resources and time, we plan to explore this aspect in future research.
>
> **W, Q1.** To verify DWD’s scalability to larger datasets, we conducted additional experiments on the ImageNet-100 dataset, which is larger than ImageNet-30. To construct the ImageNet-100 dataset, we sub-sampled 100 classes from ImageNet, as described in [D]. We divided these classes equally into 50% ID and 50% OOD classes, following alphabetical order. From each ID class, we selected a small portion (10% or 30%) as labeled data, with the remaining data forming the unlabeled set. We report the results in Table B in the PDF.
>
> As shown in the table, DWD-SL outperforms the baselines, and DWD-UT significantly enhances baseline performance. From these results, we can conclude that DWD can be applied effectively to larger datasets.
>
> **Q2.** Our training scheme, which uses the discriminator to obtain importance weights for unlabeled data, can be applied to other generative models since such importance sampling does not impose any specific restrictions on the types of generative models. However, since the proposed data generation method starts from a partially noised unlabeled image, it cannot be applied to generative models whose generation process does not involve a noisy image as an intermediate step.
>
> **Q3.** We acknowledge the importance of validating generation performance. We therefore measured the FID and IS scores of the generated samples. Additionally, we included intra-cluster pairwise LPIPS distance [E] which represents the degree of overfitting, as FID and IS scores might not capture overfitting issues in domains with limited data. To investigate whether DWD improves generation performance, we assessed these three scores under different training schemes: (a) with labeled data only, (b) with both labeled and unlabeled data using the objective (6) in our paper, and (c) with DWD.
>
> |                           | FID score (↓) | IS score (↑) | LPIPS (↑) |
> |---------------------------|---------------|--------------|-----------|
> | (a) Labeled data only     | 60.1          | 21.4         | 0.167     |
> | (b) Labeled and unlabeled | 69.2          | 20.4         | 0.170     |
> | **(c) DWD**               | **43.6**      | **23.9**     | **0.177** |
>
> We observed that case (b) results in a better intra-cluster pairwise LPIPS distance but worse FID and IS scores compared to case (a). This implies that while utilizing unlabeled data helps mitigate overfitting problems, irrelevant unlabeled data makes it difficult for the diffusion model to learn the distribution of the labeled data. On the other hand, DWD achieves the best scores across all three metrics. This indicates that DWD effectively addresses both overfitting issues and the problems caused by irrelevant unlabeled data, leading to better generation performance.
>
> We appreciate the suggestions for more diverse validation of our method, and hope that the response above addresses your comments.
>
>
> **References**
>
>
> [A] You et al., “Diffusion models and semi-supervised learners benefit mutually with few labels”, NeurIPS, 2023.
> [B] Bao et al., “All are worth words: a vit backbone for score-based diffusion models”, NeurIPS Workshop on Score-Based Methods, 2022.
> [C] Peebles et al., “Scalable Diffusion Models with Transformers”, ICCV, 2023.
> [D] K Cao et al., “OPEN-WORLD SEMI-SUPERVISED LEARNING.”, ICLR, 2022.
> [E] Ojha et al., “Few-shot Image Generation via Cross-domain Correspondence”, CVPR, 2021.

---

> > ### Comment · Reviewer_9H8V · 2024-08-10
> >
> > Thank you for your response. This addresses my main concerns. I am now inclined to accept this paper and have increased my score to a weak accept. I look forward to seeing the final version. good luck!

---

> > > ### Author Response · Authors · 2024-08-14
> > > **Official Comment by Authors**
> > >
> > > We're pleased to know that your concerns have been resolved.
> > > We will integrate all the points we talked about with you into the updated manuscript.

---

### Official Review · Reviewer_KfDe · 2024-07-11

**Soundness:** 3
**Presentation:** 3
**Contribution:** 3
**Rating:** 6
**Confidence:** 4

**Summary:**

This paper proposes DWD, a new OSSL method that trains a diffusion model to transform OOD unlabeled data to ID images for SSL. DWD can mitigate the class mismatch problem in the OSSL task, which affects SSL performance.

**Strengths:**

While previous OSSL methods attempted to distinguish between ID and OOD data through OOD detection training methods, DWD offers a novel perspective by addressing the OSSL problem from the data generation level, which leads to the state-of-the-art OSSL performance.

**Weaknesses:**

1. The experimental results are expected to be more extensive, for instance, using 1) different ratios of ID and OOD classes and 2) different ratios of labeled and unlabeled data on the same dataset to verify the method’s effectiveness. Such settings are common in recent OSSL methods like [1,2]

2. DWD does not truly enable the model to distinguish between ID and OOD data; instead, it eliminates the OOD samples from the data. Thus, DWD cannot handle potential unseen OOD samples in real-world scenarios.

3. DWD relies on the ratio of OODs in the unlabeled dataset as the prior knowledge to train the discriminator. Such prior knowledge is not always available in real-world scenarios.

[1] Li Z, Qi L, Shi Y, et al. Iomatch: Simplifying open-set semi-supervised learning with joint inliers and outliers utilization, ICCV 2023

[2] Saito K, Kim D, Saenko K. Openmatch: Open-set consistency regularization for semi-supervised learning with outliers, NeurIPS 2021

**Questions:**

For the two usage scenarios of generated data in Sec 4.2, the authors claim that the performance is better when it is served as unlabeled data. This slightly contradicts my intuition. If the generated data is reliable, why does removing labels for training result in better performance?

**Limitations:**

Please refer to the weakness.

---

> ### Author Rebuttal · Authors · 2024-08-07
>
> We appreciate your thoughtful feedback.
>
> **W1.** We agree that a more extensive set of experiments can help validate DWD’s effectiveness. We thus conducted additional experiments that are common in recent OSSL methods: 1) different ratios of ID and OOD classes, and 2) various sizes of labeled data.
>
> 1) Different ratios of ID and OOD classes
>
> As shown in Figure 3 of our paper, we conducted experiments on the SixAnimal task with different ratios of ID and OOD classes ($\zeta$ = 50%, 75%, 100%). These experiments demonstrated DWD's ability to enhance baseline SSL performance, under various class distribution mismatch scenarios. To further expand the results, we conducted additional experiments to include  $\zeta$=25% (below 50%). We report the results in Table A in PDF.
>
> Since DWD is designed to resolve the class distribution mismatch, its effectiveness is expected to decrease with a lower mismatch ratio. However, we observed that DWD was still able to improve the baseline method in the 𝛇 = 25% case, although the performance gain was relatively diminished.
>
> 2) Varying the size of labeled data
>
> To provide a more extensive set of experiments, we conducted additional experiments on the ImageNet-100 dataset, varying the size of labeled data. To build the ImageNet-100 dataset, we sub-sampled 100 classes from ImageNet, as described in [A]. We divided these classes equally into 50% ID and 50% OOD classes following alphabetical order. From each ID class, we selected a small portion (10% or 30%) as labeled data with the remaining data forming the unlabeled set. We report the results in Table B in the PDF.
>
> As shown in the table, DWD still proves to be effective with varying amounts of labeled data, performing well with both 10% and 30% sampling ratios. Notably, the performance gain is greater with the 10% sampling ratio compared to the 30% ratio, as the benefit of data augmentation is more pronounced when the dataset is smaller. However, it is important to note that DWD’s effectiveness is expected to be marginal when the amount of labeled data is extremely small, as the diffusion model struggles to accurately capture the labeled data distribution. We remark that this limitation is also common to other generative augmentation approaches.
>
>
> **W2.** Although we didn’t explicitly handle OOD samples in the paper, it is important to note that our approach involves training a separate discriminator to distinguish between ID and OOD data during the diffusion model training. By integrating this discriminator with the trained classifier, we can straightforwardly reject OOD samples and only classify ID samples during inference time.
>
>
> **W3.** We remark that we did not assume any prior knowledge of $\mu$ in our experiments. Instead, we treated $\mu$ as a hyper-parameter and systematically tuned it using a validation set, following the standard protocol for hyper-parameter tuning in the SSL literature.
>
>
> **Q1.** We understand the possible ambiguity introduced by the last sentence in Section 4.2. The sentence can be interpreted either way: (1) DWD-UT demonstrates better results than DWD-SL and/or (2) applying DWD-UT to baseline SSL methods improves the baseline performance. We intended to claim the latter rather than the former interpretation. We will carefully rephrase the corresponding sentence to mitigate this ambiguity.
>
> Still, as discussed in Section 5.2, we found that utilizing the DWD-UT with baseline SSL methods outperformed DWD-SL in some cases. We suspect this is because there are variations in the qualities of the generated images, since they are generated without considering the relevance between the class condition and the guide image. If the class condition is highly relevant to the guide image (e.g. very similar class), a high-quality image would be generated; otherwise, the quality of the generated image can be relatively low. When DWD-UT is combined with sophisticated data selection mechanisms commonly present in SSL methods (e.g., thresholding used in pseudo-labeling, weighting functions in filtering-based methods), it can discern high versus low quality samples, thereby enhancing performance compared to DWD-SL.
>
> **Reference**
>
> [A] K Cao et al., “OPEN-WORLD SEMI-SUPERVISED LEARNING.”, ICLR, 2022.

---

> > ### Comment · Reviewer_KfDe · 2024-08-12
> >
> > I appreciate the author's rebuttal and the efforts made to address my concern. I have no other problems.

---

> ### Author Response · Authors · 2024-08-14
> **Official Comment by Authors**
>
> We're glad to hear that your concerns have been addressed, and we sincerely appreciate your positive review!
> We will incorporate all the findings discussed with you into the revised manuscript.

---

### Author Rebuttal · Authors · 2024-08-07

## **General response**

We appreciate all the reviewers for taking the time to provide constructive feedbacks on our paper. We are very encouraged that the reviewers have recognized the following strengths in our work:
1) Proposition of a novel perspective by addressing the class distribution mismatch problem at the data generation level. (Reviewer KfDe, 5EHv)
2) Presentation of a creative methodology with clear motivation. (Reviewer 5EHv, 9H8V)
3) Demonstration of strong empirical performance. (Reviewer KfDe, 9H8V, 5EHv)
4) Provision of effective visualizations that aid in understanding. (Reviewer 5EHv)
5) Inclusion of comprehensive analysis with additional experiments in the appendix. (Reviewer 9H8V)

Below, we summarize the main concerns raised:
1) Necessity of assessing DWD on larger datasets. (Reviewer 9H8V, 5EHv)
2) Necessity of evaluating DWD with different ratios of ID and OOD classes and varying sizes of labeled data. (Reviewer KfDE, 5EHv)
3) Lack of analysis on hyper-parameter sensitivity, generation performance, and trade-offs between performance gains and computational cost. (Reviewer 9H8V, 5EHv)

We have addressed the comments with individual responses. If you have any questions or require further clarification, please let us know, and we will be glad to address them during the discussion period.

Additionally, please refer to our one-page PDF, which includes the following results:
* Table A : SixAnimal task with different ratios of ID and OOD classes.  (Reviewer KfDE, 5EHv)
* Table B : ImageNet-100 task with varying sizes of labeled data. (Reviewer KfDE, 9H8V, 5EHv)

---

### Decision · Program_Chairs · 2024-09-25

**Decision:**

Accept (poster)

**Comment:**

This paper focuses on the problem of open-set semi-supervised learning, where there is a large number of irrelevant instances in the unlabeled data that do not belong to any class in the labeled data. To solve the open-set semi-supervised learning problem, this paper proposes a data-centric generative augmentation approach that leverages a diffusion model to enrich labeled data using both labeled and unlabeled samples. Comprehensive experimental results demonstrate the effectiveness of the proposed approach.

After the authors' rebuttal, all the reviewers gave positive ratings to this paper (i.e., 3 x Weak Acceptance) and recognized that the proposed approach is novel and the performance is satisfactory. Therefore, I recommend accepting this paper as a poster presentation.

Please include the additional experimental results of the SixAnimal task and the ImageNet-100 task in the final version to further improve this paper.